# Validation of *Pf*SNP-LAMP-Lateral Flow Dipstick for Detection of Single Nucleotide Polymorphism Associated with Pyrimethamine Resistance in *Plasmodium falciparum*

**DOI:** 10.3390/diagnostics10110948

**Published:** 2020-11-13

**Authors:** Suganya Yongkiettrakul, Fassou René Kolié, Darin Kongkasuriyachai, Jetsumon Sattabongkot, Wang Nguitragool, Namfon Nawattanapaibool, Chayanut Suansomjit, Saradee Warit, Niwat Kangwanrangsan, Sureemas Buates

**Affiliations:** 1National Center for Genetic Engineering and Biotechnology (BIOTEC), National Science and Technology Development Agency (NSTDA), Pathum Thani 12120, Thailand; suganya.yon@biotec.or.th (S.Y.); darin@biotec.or.th (D.K.); saradee@biotec.or.th (S.W.); 2Department of Microbiology, Faculty of Science, Mahidol University, Bangkok 10400, Thailand; fassourenek@gmail.com (F.R.K.); namfon.naw@gmail.com (N.N.); 3Mahidol Vivax Research Unit, Faculty of Tropical Medicine, Mahidol University, Bangkok 10400, Thailand; jetsumon.pra@mahidol.ac.th (J.S.); chayanut_titti@outlook.com (C.S.); 4Department of Molecular Tropical Medicine and Genetics, Faculty of Tropical Medicine, Mahidol University, Bangkok 10400, Thailand; wang.ngu@mahidol.edu; 5Department of Pathobiology, Faculty of Science, Mahidol University, Bangkok 10400, Thailand; niwat.kan@mahidol.ac.th

**Keywords:** dihydrofolate reductase, loop-mediated isothermal amplification-lateral flow dipstick (LAMP-LFD), malaria detection, *Plasmodium falciparum*, pyrimethamine, antifolate resistance, drug resistance, single nucleotide polymorphism

## Abstract

The loop-mediated isothermal amplification coupled with lateral flow dipstick (*Pf*SNP-LAMP-LFD) was recently developed to detect single nucleotide polymorphism (A**A**T → A**T**T), corresponding to substitution of asparagine to isoleucine at amino acid position 51 in the *P. falciparum*
*dhfr-ts* gene associated with antifolate resistance. In this present study, the *Pf*SNP-LAMP-LFD was validated on 128 clinical malaria samples of broad ranged parasite densities (10 to 87,634 parasites per microliter of blood). The results showed 100% accuracy for the detection of single nucleotide polymorphism for N51I mutation. Indeed, the high prevalence of N51I in the *Pfdhfr-ts* gene detected in the clinical samples is in line with reports of widespread antifolate resistant *P. falciparum* in Thailand. The relationship between enzyme choice and reaction time was observed to have an effect on *Pf*SNP-LAMP-LFD specificity; however, the method yielded consistent results once the conditions have been optimized. The results demonstrate that *Pf*SNP-LAMP-LFD is a simple method with sufficient sensitivity and specificity to be deployed in routine surveillance of antifolate resistance molecular marker and inform antimalarial management policy.

## 1. Introduction

The global malaria morbidity and mortality have decreased substantially over the past decades, yet malaria still remains a significant global health threat. More than 228 million malaria cases and 405,000 deaths were reported in 2019 [1]. The increasing threats of multidrug-resistant malaria have raised the urgency to accelerate the global malaria elimination agenda. The Greater Mekong Subregion is the hotspot for multidrug-resistant malaria, where resistant parasites have emerged and spread across the globe. The sulfadoxine-pyrimethamine (SP) combination is recommended by the World Health Organization (WHO) to be used as seasonal malaria chemoprevention (SMC) in children under 5 years and as an intermittent preventive therapy (IPT) in pregnant women in areas of moderate or high transmission of *Plasmodium falciparum* [2,3]. SP blocks the enzymes in the folate synthesis pathway of *P. falciparum*, the dihydropteroate synthetase (DHPS), and the dihydrofolate reductase (DHFR), respectively [4,5]. Pyrimethamine resistance arose from specific point mutations resulting in amino acid substitutions in the DHFR at positions N51I, C59R, S108N/T, and I164L [6,7,8,9,10,11,12].

The surveillance of molecular markers associated with antimalarial resistance can be informative for evidence-based policy decision on antimalarial management. The prevalence of known single nucleotide polymorphisms (SNPs) attributed to pyrimethamine resistance was considered during the development of SNP-loop-mediated isothermal amplification coupled with lateral flow dipstick (*Pf*SNP-LAMP-LFD). While S108N is the most common SNP identified; however, single mutation is rarely found in pyrimethamine resistant parasites [13,14,15]. In fact, stepwise accumulation of *Pfdhfr* has been described with an initial S108N mutation and sequential additional mutations at N51I and/or C59R, and I164L [15]. The high A + T content of *P. falciparum* genome represented technical challenges in the design of a SNP-LAMP primer set. The position of N51I SNP allowed for the best primer design and development of a robust *Pf*SNP-LAMP-LFD that could capture most of the circulating pyrimethamine-resistant parasite population including double mutant (N51I + S108N), triple mutant (N51I + C59R + S108N), and quadruple mutant (N51I + C59R + S108N + I164L).

Loop-mediated isothermal amplification (LAMP) is a nucleic acid amplification method performed under single temperature that is relatively simple, cost-effective, and time-efficient [16]. Key features of LAMP are the proper design of four different primers that bind to six different regions of the target sequence and the use of *Bst* DNA polymerase with strong strand displacement activity which eliminate the need for double strand denaturing steps compared to traditional PCR-based method. The use of four primers in LAMP method can quickly accumulate a large amount of LAMP products with characteristic stem-loop concatemeric structures and pyrophosphate by-products, which can form visible white precipitants in the presence of magnesium. Commonly used methods for visualizing LAMP products include gel electrophoresis, real-time turbidimetry, SYBR green dye, and a lateral flow dipstick (LFD). The use of gel electrophoresis can distinguish specific amplification from non-specific products but it is more time consuming, while the visualization of SYBR green dye intercalated LAMP products is simple but can be subjective. Real-time turbidity resulting from precipitation of magnesium-pyrophosphate can be used; however, it requires the purchase and maintenance of a turbidimeter. The LAMP method does not require specialized equipment, so it has the potential to be used in molecular diagnostic at point-of-care or in surveillance programs. Indeed, LAMP test kits are available for the detection variety of pathogens such as COVID-19, *E. coli* O157, and tuberculosis [17].

There have been numerous reports demonstrating comparable or even improved performances of LAMP-based protocols compared to PCR-based protocols for species detection of malaria parasites [18,19,20,21]. In recent years, several LAMP-based single nucleotide polymorphism (SNP) detection protocols with different result readouts have been reported. LAMP coupled one-step strand displacement (LAMP-OSD), and LAMP combined with allele-selective oligonucleotide hybridization (LAMP-ASO), relied on the use of allele-specific probe hybridization to detect amplified LAMP products [22,23]. Probe-enhanced LAMP (PE-LAMP) utilized the loop-primer as allele-specific (AS) primer that bound to target allele or SNP and preferentially amplified the target sequence, while the LAMP-SNP utilized the FIP and BIP inner primers as sequence-specific primers to preferentially amplify the target sequence [24,25,26]. Peptide nucleic acid-locked nucleic acid-mediated LAMP (PNA-LNA LAMP) utilized peptide nucleic acid as a blocking agent to prevent the amplification of a specific allele or SNP [27]. Compared to other LAMP-AS or LAMP-SNP protocols, which relied on colorimetric change observed by naked eyes or in real-time PCR equipment, the *Pf*SNP-LAMP-LFD used the same read out format as most commercially available rapid diagnostic tests to detect N51I mutation in the *Pfdhfr-ts* gene [26]. In this present study, the *Pf*SNP-LAMP-LFD was further optimized and evaluated on clinical malaria samples, in order to validate its application as an alternative test for point-of-care diagnostics and for the molecular surveillance of malaria drug-resistant biomarkers.

## 2. Materials and Methods

### 2.1. Clinical Blood Sample Collection and Genomic DNA Extraction

Blood samples were collected from malaria patients presented at the malaria clinics in Thailand between 2013 and 2018. The prevalence of *P. falciparum* in this area was approximately 3.7%, as determined by PCR [28]. Patients were diagnosed with malaria based on microscopy examination of thick and thin blood films by laboratory technicians and received the standard treatment according to the national treatment guidelines. Patients with uncomplicated *P. falciparum* infection were invited to participate in the study after providing informed consent, following an approved protocol by the Ethics Committee of the Faculty of Tropical Medicine, Mahidol University (Protocol number: TMEC 11-030, TMEC 16-010, and TMEC 18-009). Briefly, blood sample was collected from each patient by finger prick using heparinized capillary tubes. A volume of 100 µL of the packed red blood cells was stored in the freezer. Genomic DNA extraction was performed on the frozen blood samples using ^®^QIAamp DNA Blood minikit (Germantown, MD, USA), as described by the manufacturer’s protocol. A total of 2 µL of eluted DNA was used directly for *Pf*SNP-LAMP-LFD to detect N51I SNP on the *dhfr-ts* gene associated with pyrimethamine resistance and for nested PCR to confirm malaria species based on the detection of the 18S ribosomal RNA gene [29].

### 2.2. DNA Sequencing

Briefly, the amplified *Pfdhfr*-*ts* gene products of *P. falciparum* clinical samples were obtained by PCR using specific primer pairs (Forward-*Pfdhfr*; 5′-GATGGAACAAGTCTGCGACGTTTTCG-3′ and Reverse-*Pfdhfr*; 5′-CCCAAGTAAAACGATTAGATCTTCAACTTT-3′). PCR reactions were conducted in a 25 µL reaction mixture using the following condition: initial denaturation at 95 °C for 2 min, followed by 35 cycles of 95 °C for 1 min, 51 °C for 1 min, and 72 °C for 1 min, with a final extension at 72 °C for 5 min. A PCR purification kit (NIPPON Genetics, Tokyo, Japan) was used to obtain purified PCR products for DNA sequencing. The sequencing reactions were conducted with the BigDye Terminator Version 3.1 Cycle-Sequencing Kit (Applied Biosystems, Foster City, CA, USA), using the same primers as above. The DNA sequences were analyzed with the Bioedit program.

### 2.3. Recombinant Plasmid Construction

The pUC18-*Pfdhfr*-TM4/8.2 containing the wild-type SNP (AAT) of the *Pfdhfr-ts* gene was used as a negative control. The pUC18-*Pfdhfr*-V1/S containing the mutant SNP (ATT) corresponding to the N51I mutation was used as positive control. The V1/S strain of *P. falciparum* is considered to be highly resistant to pyrimethamine, with 4 reported mutations (N51I + C59R + S108N + I164L) in the *Pfdhfr-ts* gene. These two plasmids were constructed as previously described [21].

### 2.4. PfSNP-LAMP-LFD Conditions

The 25 µL-volume of *Pf*SNP-LAMP reaction mixture contained the following components: 2 µM each of *Pf*-snp-FIP and *Pf*-snp-BIP primers, 0.2 µM each of *Pf*-snp-F3 and *Pf*-snp-B3 primers, 1X isothermal amplification buffer (20 mM Tris-HCl, 10 mM (NH_4_)_2_SO_4_, 10 mM KCl_2_, 2 mM MgSO_4_, 0.1% Triton X-100, pH 8.8), 0.4 M betaine (USB Corporation, Cleveland, OH, USA), 8 mM MgSO_4_ (Sigma-Aldrich, St. Louis, MO, USA), 1.4 mM dNTP mix (Promega, Madison, WI, USA), 8 units of *Bst* DNA polymerase, a large fragment or *Bst* 2.0 DNA polymerase or *Bst* 2.0 WarmStart DNA polymerase (New England Biolab, Ipswich, MA, USA), and 2 µL of DNA sample. The design of the *Pf*SNP-LAMP primer set was also previously described [26]. For negative and positive control reactions, 20 nanograms of pUC18-*Pfdhfr*-TM4/8.2 and pUC18-*Pfdhfr*-V1/S were used, respectively. For optimization, LAMP reactions were performed at 60–63 °C and observed for 60–90 min and LAMP product signals were monitored using Loopamp Realtime Turbidimeter at 650 nm wavelength (LA-320C, Eiken Chemical Co., Ltd., Tokyo, Japan). For the validation of clinical samples, *Pf*SNP-LAMP reactions were performed using *Bst* 2.0 WarmStart DNA polymerase (New England Biolab, Ipswich, MA, USA) at 63 °C for 75 min. Then, 20 pmol *Pf*-FITC-probes was directly added to the reaction and allowed to hybridize with products at 63°C for 5 min. Subsequently, 8 µL of the hybridized *Pf*SNP-LAMP-probe were transferred into a new Eppendorf tube containing 120 µL of room-temperature assay buffer (Milenia^®^ GenLine HybriDetect, GieBen, Germany). The LFD strip was dipped into the assay solution for 5 min to allow the solution to migrate by chromatography effect. Two bands were observed on the control line and the test line of the LFD strip for samples with N51I mutation. One band was observed on the control line, representing a negative result for negative sample (no DNA template) or wild-type SNP sample. If no signal appeared on the control line, then the result was invalid.

## 3. Results

### 3.1. Effects of Enzyme and Reaction Time on PfSNP-LAMP Sensitivity and Specificity

*Bst* 2.0 DNA polymerase and *Bst* 2.0 WarmStart DNA polymerase are homologues of *Bst* DNA polymerase large fragment that have been designed for improvements in amplification speed, thermostability, and salt tolerance, in order to increase the quantity of product, according to the manufacturer’s product descriptions. Here, we compared the effect of enzyme choice on *Pf*SNP-LAMP reactions. The *Pf*SNP-LAMP reactions were evaluated under varying temperatures from 60 to 63 °C for up to 90 min using Loopamp Realtime Turbidimeter.

Figure 1A–D show the effects of temperature, choice of enzyme, and length of observed reaction time, on the sensitivity and specificity of *Pf*SNP-LAMP. Signals represented amplified products using *Bst*, *Bst* 2.0, and *Bst* 2.0 WarmStart DNA polymerases in the reactions with either pUC18-*Pfdhfr*-V1/S (Lines 1, 3, and 5) or pUC18-*Pfdhfr*-TM4/8.2 (Lines 2, 4, and 6) as DNA templates.

The overall observation suggested that all three versions of *Bst* DNA polymerases could distinguish the V1/S SNP (ATT, isoleucine) from TM4/8.2 SNP (AAT, asparagine) when performed at the optimal temperature of 63 °C. Signals began to appear around the 60-min mark and reached the optimal time around the 75-min mark. The appearance time of peak signal was dependent on the amount of the starting DNA templates in the reaction. The signal intensities were similar for all reaction conditions and concentrations of templates tested.

Non-specific signals could be observed from samples with pUC18-*Pfdhfr*-TM4/8.2 when using *Bst* DNA polymerase (Line 2) for all temperatures. Non-specific signals clustered closer to the true positive signals (Lines, 1, 3, and 5, near the 60-min mark) when performed at the lower range of temperature at 60 °C (Figure 1A) and 61 °C (Figure 1B), thus, reducing the cut-off window for distinguishing between non-specific signals from true positive signals. There were greater separations between true positive signals (Lines 1, 3, and 5, below the 60-min cut-off mark) and non-specific signals (Lines 2 and 4, above the 60-min cut-off mark) when reactions were performed at 62 and 63 °C as shown in Figure 1C and Figure 1D, respectively. The optimal reaction temperature and enzyme combination to yield the best separation of true positive signals, and non-specific signals were observed at 63 °C for *Bst* 2.0 WarmStart DNA polymerase, with a reaction time window between 60–75 min.

To further explore the detection limit of *Pf*SNP-LAMP reaction with *Bst* 2.0 WarmStart DNA polymerase, reactions were performed at 63 °C by using ten-fold serial dilutions of DNA templates from 0.002 to 200 ng. No signal was observed from pUC18-*Pfdhfr*-TM4/8.2 up to the 75-min mark, as expected for a negative control. However, a non-specific signal could be observed with 200 ng of pUC18-*Pfdhfr*-TM4/8.2 above the 77-min mark, which was beyond the optimal range of reading time between 60–75 min (Figure 2A). The *Pf*SNP-LAMP showed a dose response effect for detection of mutant SNP, where samples with higher concentration of pUC18-*Pfdhfr*-V1/S showed a faster appearance of peak signals, corresponding to accumulation of amplified products (Figure 2B), as expected. The detection limit for *Pf*SNP-LAMP reaction using *Bst* 2.0 WarmStart DNA polymerase appeared to be 0.02 ng of pUC18-*Pfdhfr*-V1/S (~5.5 × 10^5^ copy number). These results were consistent with the detection limit of pUC18-*Pfdhfr*-V1/S observed in previous report [26]. Furthermore, the *Pf*SNP-LAMP reaction products were resolved on gel electrophoresis where the characteristic ladder pattern was observed for reactions amplified from pUC18-*Pfdhfr*-V1/S but not from pUC18-*Pfdhfr*-TM4/8.2 (data not shown). The amplified *Pf*SNP-LAMP products were visualized on LFD strips, as shown in Figure 3.

### 3.2. Validation of PfSNP-LAMP-LFD in Clinical Blood Samples from Malaria Patients

A total of 128 clinical blood samples from malaria clinics in Thailand were used to validate *Pf*SNP-LAMP-LFD. There were 55 *P. falciparum* samples and 73 *P. vivax* samples, the latter served as additional negative control samples. A recent investigation of *P. falciparum* samples collected in Thailand between 2008–2016 showed high prevalence of N51I among the surveyed parasites, where up to 11% were triple mutants (N51I + C59R + S108N) and 83% were quadruple mutants (N51I + C59R + S108N + I164L) [30,31]. Due to limited availability of clinical samples with wild-type *Pfdhfr-ts* in Thailand, we included 20 ng of genomic DNA from two different lab strains, with wild-type *Pfdhfr-ts* (TM4/8.2 and NF54) as negative controls for this validation study.

The validation results are summarized in Table 1. The *Pf*SNP-LAMP-LFD reactions correctly detected SNP associated with pyrimethamine resistance (ATT, N51I) in 55 out of 55 *P. falciparum* samples, with parasite density from 10 to 87,634 parasites per microliter (P/µL) of blood (100% accuracy), while no signal was observed in all 73 *P. vivax* samples with parasite density from 9 to 12,632 P/µL and *P. falciparum* strain NF54 sample (100% specificity). The range of parasite density used in this study was sufficiently broad to allow insights into the performance of *Pf*SNP-LAMP-LFD in actual clinical settings. The performance observed for *Pf*SNP-LAMP-LFD was comparable to the typical performance range seen with expert level malaria microscopy (50–500 P/µL), commercially available rapid diagnostic test (RDT, 100–200 P/µL), and other nucleic acid-based detection methods (1–5 P/µL) [32,33,34]. DNA sequencing confirmed that all of the 55 *P. falciparum* samples contained the mutant SNP in the *Pfdhfr-ts* gene. In addition, C59R and S108N mutations were also observed in these samples; however, it is noted that SNP associated with I164L in the *Pfdhfr-ts* gene was not examined. Nonetheless, these observations were in agreement with other studies that also reported high prevalence of N51I, C59R, S108N, and I164L mutations in the *Pfdhfr-ts* gene, among circulating parasites in the Greater Mekong Subregion [30,31]. Representative samples of *Pf*SNP-LAMP-LFD validation on clinical malaria samples tested are shown in Figure 4.

## 4. Discussion

The primary objective of this study was to validate the *Pf*SNP-LAMP-LFD using clinical samples while exploring the robustness of the method when performed under different conditions. Our results demonstrated that the *Pf*SNP-LAMP-LFD could accurately distinguish SNP (AAT → ATT, N51I) associated with pyrimethamine resistance on 128 clinical samples with a broad range of parasite density from as low as 10 parasites/µL of blood. The *Pf*SNP-LAMP-LFD performance was comparable to typical LAMP-based protocols and other PCR-based protocols for distinguishing *Plasmodium* species [35]. For a more direct comparison, a recent publication by Chahar et al. reported a protocol for SNP-LAMP with hydroxynapthol blue indicator for the detection of S108N mutation in the *Pfdhfr-ts* gene with the detection limit of seven parasites/µL at 60 °C for 45 min [36].

In general, we noted that *Pf*SNP-LAMP-LFD method, once optimized, was sufficiently robust to withstand some variations to the protocol, such as batch-to-batch variation of reagents, equipment and laboratory environment, and the experience level of laboratory technicians. Nonetheless, we observed that the efficiency of *Pf*SNP-LAMP-LFD to distinguish mutant SNP from wild-type SNP was affected by the relationship between enzyme choice and the reaction cut-off time. In our observations at 63 °C, amplified signals from mutant samples could be detected within the 60-min mark for all versions of *Bst* DNA polymerases, with *Bst* DNA polymerase large fragment showing the earlier peak signal, followed by *Bst* 2.0 DNA polymerase and *Bst* 2.0 WarmStart DNA polymerase. However, non-specific signals were detected in wild-type samples when the reactions extended well beyond the 60-min mark. We observed an approximate delay of 25 min between the appearance of peak signals from mutant samples and wild-type samples for all three versions of the *Bst* DNA polymerases used in the study; this delay represented the window cut-off which *Pf*SNP-LAMP-LFD could distinguish between the mutant and wild-type *Pfdhfr-ts* gene at codon 51. In previous *Pf*LAMP and *Pf*SNP-LAMP development efforts, the amplification reactions were optimized within the 60-min mark, in favor of a faster turnaround of reliable results; therefore, we did not observe the non-specific signals in prior experiments [21,26].

Several factors can be attributed to non-specific amplification in LAMP products, including sample quality and partial hybridization of one or more LAMP primers to the fragmented DNA of target organisms or of host DNA. To ensure sample quality, purified genomic DNA samples were used for *Pf*SNP-LAMP-LFD, and efforts were made to not exceed more than two freeze-thaw cycles of DNA samples and reagents. The design of the *Pf*SNP-LAMP primer set relies on a single nucleotide base difference to distinguish between mutant from wild-type genotypes; the efficiency and stability of primers binding to the mutant sequences would be more favorable than binding to the wild-type sequences. This preferential binding can be leveraged by optimizing the relationship between enzyme and reaction time to promote favorable amplification from mutant sequences over wild-type sequences, thereby improving the sensitivity and specificity of *Pf*SNP-LAMP. Similar observation patterns were made on the effects of enzyme choice and reaction temperature on the sensitivity and specificity of SNP-LAMP by Mohon et al. in their development of SNP-LAMP to detect the C580Y mutation in the *Kelch13* gene corresponding to artemisinin-resistance in *P. falciparum* [37]. These observations further highlighted the key components to consider for successful development of SNP-LAMP, which included primer designs, enzyme choice, and cut-off reaction time.

The current *Pf*SNP-LAMP-LFD protocol includes a genomic DNA purification step, and thus requires the laboratories to be equipped with basic equipment, including microcentrifuge, temperature-control heating block or water bath, and pipetting instruments. Simplified sample preparation can improve the applicability of nucleic acid-based detection method as point-of-care diagnostics, particularly for use in more remote settings where malaria is endemic. However, this convenience in sample preparation can affect the quality of starting DNA templates, and consequently the limit of detection. Previous attempts to simplify DNA preparation showed a 10-fold difference in the limit of detection of *Pf*SNP-LAMP-LFD, for example, when purified genomic DNA (LOD = 0.2 ng of genomic DNA) was compared to non-purified genomic DNA (LOD = 2 ng of genomic DNA in the presence of red blood cell lysate) [26]. The overall results and previous observations suggest that *Pf*SNP-LAMP-LFD can provide consistent results for the detection of SNP associated with pyrimethamine resistance, which can be a proxy for treatment efficacy, particularly in areas where a sulfadoxine-pyrimethamine regimen is still in use. With increasing threats of multidrug-resistant malaria, *Pf*SNP-LAMP-LFD can be used as part of a toolkit for a comprehensive molecular surveillance program to monitor known biomarkers for drug resistance to support evidence-based malaria treatment policies.

## Figures and Tables

**Figure 1 diagnostics-10-00948-f001:**
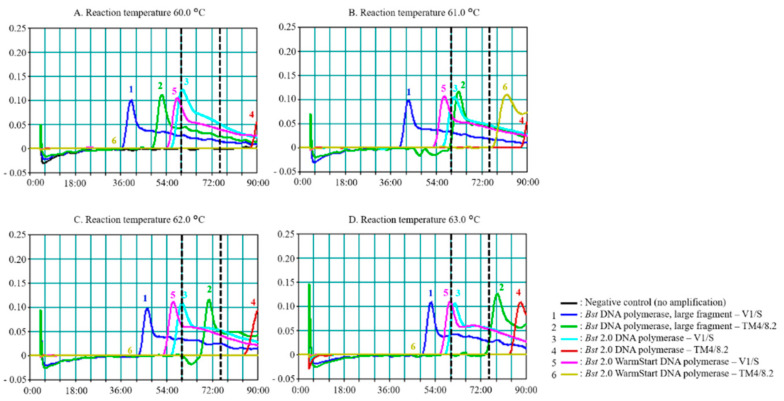
Effects of temperature and enzyme choice on SNP-loop-mediated isothermal amplification (*Pf*SNP-LAMP) performance to detect mutant single nucleotide polymorphism (SNP) in the *Pfdhfr-ts* gene. Reactions were performed at (**A**) 60.0 °C, (**B**) 61.0 °C, (**C**) 62.0 °C, and (**D**) 63.0 °C. A total of 20 ng of template DNA was used for each reaction. Lines 1 and 2 were reactions using *Bst* DNA polymerase large fragment on pUC18-*Pfdhfr*-V1/S (mutant SNP) and pUC18-*Pfdhfr*-TM4/8.2 (wild-type SNP), respectively. Lines 3 and 4 were reactions using *Bst* 2.0 DNA polymerase on V1/S and TM4/8.2, respectively. Lines 5 and 6 were reactions using *Bst* 2.0 WarmStart DNA polymerase on V1/S and TM4/8.2, respectively. The 60-min and 75-min markers were shown with dashed lines. Signals were read by Loopamp Realtime Turbidimeter for up to 90 min. Similar results were obtained in three independent experiments.

**Figure 2 diagnostics-10-00948-f002:**
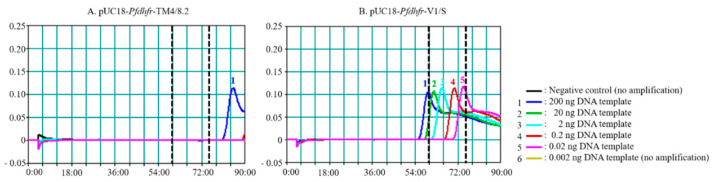
Detection limit of *Pf*SNP-LAMP reaction using *Bst* 2.0 WarmStart DNA polymerase. The amplification reactions were performed at 63 °C using 10-fold serial dilutions (0.002 to 200 ng) of (**A**) pUC18-*Pfdhfr*-TM4/8.2 and (**B**) pUC18-*Pfdhfr*-V1/S. Similar experiments were performed independently in five replications.

**Figure 3 diagnostics-10-00948-f003:**
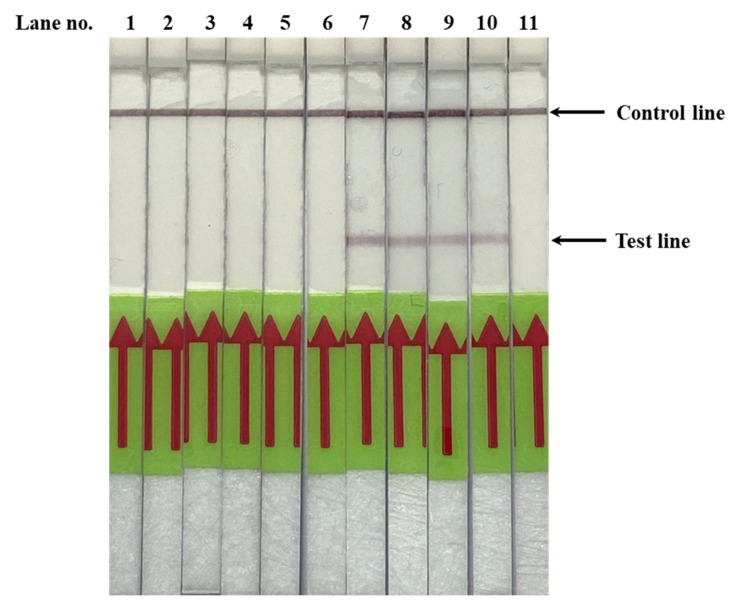
Representative results for SNP-loop-mediated isothermal amplification combined with later flow dipstick (*Pf*SNP-LAMP-LFD) assay. The products from *Pf*SNP-LAMP reactions were hybridized with labelled probes and applied to the LFD strips for visualization. Lane 1 was a negative control (no DNA template). Lanes 2–6 were pUC18-*Pfdhfr*-TM4/8.2 (wild-type SNP) diluted at 20 ng, 2 ng, 0.2 ng, 0.02 ng and 0.002 ng, respectively. Lanes 7–11 were pUC18-*Pfdhfr*-V1/S (mutant SNP) diluted at 20 ng, 2 ng, 0.2 ng, 0.02 ng, and 0.002 ng, respectively. For positive detection of the SNP mutation, both the control and test lines must appear on the strip. For wild-type SNP, only the control line appears. If no signal appears on the control line, then the result is invalid.

**Figure 4 diagnostics-10-00948-f004:**
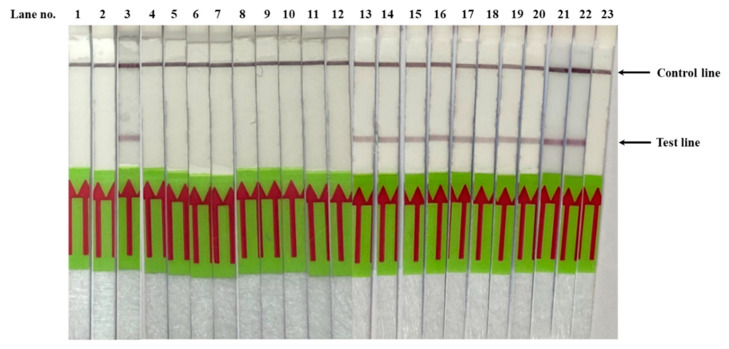
Validation of *Pf*SNP-LAMP-LFD assay performed on genomic DNA extracted from clinical samples of *P. falciparum* and *P. vivax*. Lane 1 was negative control (no DNA template). Lanes 2 and 23 were negative control samples (20 ng of pUC18-*Pfdhfr*-TM4/8.2). Lanes 3 and 22 were positive control samples (20 ng of pUC18-*Pfdhfr*-V1/S). Lanes 4–12 were genomic DNA prepared from representative *P. vivax*-infected blood samples. Lanes 13–21 were genomic DNA prepared from representative *P. falciparum-*infected blood samples with parasite density as determined by microscopy examination to be 1298 P/µL, 926 P/µL, 17 P/µL, 12,418 P/µL, 2906 P/µL, 1341 P/µL, 7886 P/µL, 2835 P/µL, and 1313 P/µL, respectively.

**Table 1 diagnostics-10-00948-t001:** Summary of results from *Pf*SNP-LAMP-LFD detection of SNP for N51I mutation in 128 clinical malaria samples from Thailand categorized by malaria species and parasite density. Samples with SNP for N51I mutation were corroborated by DNA sequencing.

	No. of Samples	(P %)	Parasite Density (P/µL) ^1^	*Pf*SNP-LAMP-LFD (N51I)
*P. falciparum* ^2^	2	ND	ND	2
	5	>0.2	>10,000	5
	25	>0.02–0.2	>1000–10,000	25
	10	>0.004–0.02	>200–1000	10
	6	>0.0002–0.004	>100–200	6
	3	0.0001–0.0002	>50–100	3
	4	<0.0001	<50	4
*P. vivax* ^3^	1	ND	ND	0
	72	<0.0002–0.24	<10–12,000	0
**TOTAL**	128			55

^1^ Expert level microscopist typical performance in the range of 50–500 P/µL blood; malaria rapid diagnostic test (RDT) typical performance in the range of 100–200 P/µL blood. ^2^ Two of 55 samples positive for *P. falciparum* had no record for parasite density (ND). ^3^ 1 of 73 samples positive for *P. vivax* had no record for parasite density (ND).

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
