# Peer review of "Validation of PfSNP-LAMP-Lateral Flow Dipstick for Detection of Single Nucleotide Polymorphism Associated with Pyrimethamine Resistance in Plasmodium falciparum"

_diagnostics, 2020, doi:10.3390/diagnostics10110948_

Round 1

Reviewer 1 Report

The manuscript presented by Yongkiettrakul and collaborators presents a validation process for a LAMP technique to detect polymorphism in clinical samples from individuals with Plasmodium falciparum, the etiological agent of malaria. The work is very well presented and the data discussed properly.

A minor suggestion:
- I think that figure 1 should be better presented. A larger size and graphic quality.

Reviewer 2 Report

In this article, Yongkiettrakul et al presents a method based on SNP-LAMP-LFD to detect a marker of antifolate pyrimethamine resistance. They present an improved method that was previously published (PMID: 27816495) by some of the author of the present article, increasing in sensitivity and validating it with clinical samples. I raise though some concerns on the primers design and the clinical samples used to validate.

In this article they have used the same primers presented in the previous reference (PMID: 27816495) which sequence on the primers Pf-snp-FIP presents a miss-difference with a nucleotide missing (underlined and bold): ATACATTTCCATGGTAATACTCCTTT-TT-CTACACATTTAGAGGTC. Unless this deletion was added on purpose to optimize the thermodynamic properties of the primer (which is not described in the text), it could affect the sensitivity of the method.

The primer B3 also sits on top of the S108N SNP, that could once more affect the sensitivity to detect the N51I of the method is used on samples that have or not a mutation at PfDHFR 108 amino acid position. The controls they have used to test the sensitivity either have the quadruple mutant (V1/S; N51I + C59R + S108N + I164L) or used the TM4/8.2 strain containing the wild-type for the 4 positions.

In this regard, the manuscript will benefit of a validation of the herein presented method with lab strains containing different combinations in regards to 51, 59 and 108 PfDHFR positions.

The same principle of validation should be considered when validating with clinical samples, although in this case, acquisition of clinical samples with different haplotypes might be very challenging.

The authors should also emphasize better in the introduction why designing a method for a N51I SNP of PfDHFR when there are other mutations (C59R + S108N + I164L) relevant for pyrimethamine resistance. There are some regions where for example the SNP S108N is more prevalent than the N51I (for example PMID: 18082233).

Reviewer 3 Report

Overall this is a relatively clear and well written manuscript presenting the optimization of an isothermal assay for the detection of the N51I SNP in the P. falciparum dhfr gene which is associated with pyrimethamine resistance. The optimized protocol is then evaluated using a relatively small panel of clinical samples of 56 P. falciparum positive samples (all carrying the dhfr N51I SNP) and 73 P. vivax samples serving as negative controls. I have a few issues that I believe can be addressed by a better presentation of the results in a revised version of the manuscript.

  • Overall: This assay detects the N51I SNP in the falciparum dhfr gene. Seeing the limited specificity of the assay in high copy number P. falciparum samples (Figure 2), it would have been nice to have seen a mixed selection of P. falciparum clinical samples (including those with high parasite density) where both the WT and mutated dhrf SNP is represented. The 73 P. vivax samples does not seem to be an adequate negative control in this assay.
  •  
  • Lines 23-24 “100% agreement with expert-level malaria microscopy for species confirmation”: This tool does not actually determine species. It detects a SNP in falciparum specifically. But if the assay is negative than this could be because it is WT Pf or another unknown species. Please revise this sentence. Also, on line 87 it states that species confirmation was done by real-time PCR, and there is nothing regarding microscopy in the methodology.
  • Lines 66-67: I suggest introducing the LAMP-LFD methodology better in the introduction. I gather that this methodology is already published, but this publication may not have been read by the audience of the target journal (Diagnostics), so perhaps a small paragraph describing how this works and what it the advantage of using the lateral flow dipstick compared to using other visual aids e.g. turbidity/visualization under UV light/hydroxynapthol blue would be good.
  • Line 159: A cut off time of 60 minutes seems really to be borderline for positivity? Anything less than 20 ng of template would have been negative for two of the enzymes at least. Please revise this statement.
  • Line 184: The limit of detection is approximately 4x10^6 copy numbers, can this be translated into an estimation of p/µL? It seems very high, yet the authors were successfully able to detect 3 p/µL with the assay in clinical samples.
  • Figure 3: I do not believe it is representative to exclude the 200ng template amplicon in Figure 3, rather I believe it important to see how the 200ng WT amplicon appears like on the LFD (i.e., is it visible? Is it very weak?).
  • Line 206: Please specify which reaction conditions and reaction times were used when assessing the clinical samples. Were these read in both the turbidimeter and LFD?
  • Line 209: Please specify the equivalent copy number/parasite density of the NF54 falciparum control.
  • Table 1: I would like to see the mean time to positivity in table 1, i.e., how long time does it take for especially the very low parasite density samples to become positive and is there any overlap with the cut-off mark for the assay (i.e., to avoid getting false positive results late in the assay)
  • Figure 4: it would be nice to know the parasite densities of the “representative” falciparum infected blood samples. I would like to see a range in parasite densities in order to see what bands look like in very low-density samples. Is there any chance of missing positive samples due to very weak bands etc.
  • Discussion: Finally, I am missing a discussion as to how this tool might be implemented. Why is this tool useful? In which settings might it be used e.g. in reference field laboratory? What is the relevance of detecting this single SNP, what does it say about the levels of drug resistance in the setting? Is this single SNP somehow representative of all 4 SNPs in dhfr?

  • Overall the English is OK, but there are a few spelling and grammar errors that need to be addressed. Some examples: lines 41 and 42 (Global malaria morbidity and mortality has….remains…); lines 58-59 “in molecular diagnostic”; line 134 “seenistivity”; line 137 “amplifcation” line 209 “genomic DNA?”

Round 2

Reviewer 2 Report

The authors have satisfactorily responded to all my comments and made the necessary changes to the manuscript.

Reviewer 3 Report

The authors have addressed my comments adequately.